# Complex Formation of Rare-Earth Elements in Carbonate–Alkaline Media

**DOI:** 10.3390/ma16083140

**Published:** 2023-04-16

**Authors:** Tatiana Litvinova, Ruslan Kashurin, Denis Lutskiy

**Affiliations:** Department of General and Physical Chemistry, Saint Petersburg Mining University, 199106 Saint-Petersburg, Russia; litvinova_te@pers.spmi.ru (T.L.); lutskiy_ds@pers.spmi.ru (D.L.)

**Keywords:** thermodynamic stability constants, rare earth elements, carbonate complexes, Gibbs energy of formation, activity coefficient

## Abstract

Rare earth metals are critical components for many industries. The extraction of rare earth metals from mineral raw materials presents many problems, both of a technological and theoretical nature. The use of man-made sources imposes strict requirements on the process. Thermodynamic and kinetic data that could describe the most detailed technological water–salt leaching and precipitation systems are insufficient. The study addresses the problem of a small amount of data on the formation and equilibrium of carbonate–alkali systems of rare earth metals. Isotherms of solubility of sparingly soluble carbonates with the formation of carbonate complexes are presented to evaluate equilibrium constants logK at zero ionic strength for Nd −11.3, Sm −8.6, Gd −8.0, and Ho −7.3. To accurately predict the system under consideration, a mathematical model was developed, which allows to calculate the water–salt composition. The initial data for calculation are concentration constants of stability of lanthanide complexes. This work will contribute to deepening knowledge about rare earth elements extraction problems and will serve as a reference for studying the thermodynamics of water–salt systems.

## 1. Introduction

The processes of extraction, processing, and treatment of rare-earth raw materials are associated with various chemical reactions such as leaching, complexation, precipitation, and solvent extraction. With the development of knowledge-intensive industries, the problem of obtaining rare earth components from different sources, both natural and man-made, has arisen and remains relevant. The technogenic sources consist of red mud, catalysts for catalytic cracking processes [1], batteries and accumulators [2], fluorescent lamps and phosphors [3,4], and magnets [5], including those which are used for wind turbines [6]. Complex treatment of the above-mentioned wastes will allow obtaining not only their useful utilization [7] but also high-margin coproducts in the form of REE compounds. The creation of such technical solutions is possible when the behavior of REE in the media of sub-acid inorganic ligands is well understood.

The works [8,9] describe prospects for recovery of REM from phosphogypsum. The integrated treatment of red mud has not been fully implemented. In general, the acid is difficult to call a versatile and complex leaching agent. The carbonate conversion of phosphogypsum is less demanding on raw materials, while it is possible to process phosphogypsum of any lying state. The potential of phosphorus feedstock for recovery of REE is described in the works [10,11]. REE in red mud are associated with iron minerals [12,13]; thus, the process is more complex.

One way of large-scale waste treatment is based on using carbonate media as directly alkali metals or ammonia carbonate solution [14,15] or as carbon dioxide in alkali media wastes [16,17]. In the case of red mud, carbon dioxide influence allows to recover scandium into an alkali carbonate solution in the form of carbonate complexes [18,19]. Yttrium and lanthanides form stable carbonate complexes of REE with the composition of $\left[Ln{CO}_{3}^{+}\right],\;\left[{Ln(CO_{3})}_{2}^{-}\right],\;and\;\left[LnHCO_{3}^{2+}\right]$ [20,21,22]. The extraction degree of lanthanides during red mud treatment is low and unstable: at similar conditions, it is possible to obtain different values of the extraction degree [23]. The unstable recovery of yttrium and lanthanides can be described by the special thermodynamic features of REE carbonate dissolution in the carbonate alkali media.

The available information on the solubility of yttrium and lanthanide compounds in carbonate media has been poorly studied. It is known that the complex formation of REE carbonate complexes occurs with an excess of carbonate ion [24,25]. In this case, an equilibrium system is formed, where multilateral equilibria with carbonate, hydroxide, and hydroxycarbonate ions should be taken into account. 

When considering the coexistence of rare earth ions with other ions of the solution, it is necessary to take into account all possible processes of complexation and precipitation of the formed compounds. The mathematical model of the precipitation–dissolution process can be built on various algorithms and contain individually selected coefficients [26,27]. The technical solution used in carbonate–alkaline leaching technology can contain various inorganic anions, which are complexing agents with REE. The following anions are most common in processing: $OH^{-}$; ${HCO}_{3}^{-}$; ${CO}_{3}^{2-}$; ${Cl}^{-}$; and ${SO}_{4}^{2-}$. The chemical reactions of carbonate complexation (1)–(3) and the formation of hydroxo-complexes (4) and (5) are significant. The complex formation occurs in several stages, with the formation of complexes of different strengths. In this regard, it is necessary to take into account all possible constants of complex stability.
(1)$${Ln}^{3+}+{CO}_{3}^{2-}↔{LnCO}_{3}^{+}$$
(2)$${Ln}^{3+}+2{CO}_{3}^{2-}↔{Ln(CO_{3})}_{2}^{-}$$
(3)$${Ln}^{3+}+{HCO}_{3}^{-}↔{Ln(HCO_{3})}^{2+}$$
(4)$${Ln}^{3+}+OH^{-}↔LnOH^{2+}$$
(5)$${Ln}^{3+}+2OH^{-}↔Ln(O{H)}_{2}^{+}$$

The heterogeneous equilibrium of the solid carbonate and solution is described by the following reaction:(6)$$Ln_{2}{\left(CO_{3}\right)}_{3\left(s\right)}↔2{Ln}_{\left(aq.\right)}^{3+}+3{CO}_{3\left(aq.\right)}^{2-}$$

A significant amount of information on the equilibrium constants of yttrium and lanthanides complexes [21,22,28,29] was obtained for a low concentrated aqueous solution in media close to neutral, for which a low ionic strength is considered. Any aqueous solution that circulates during the raw material treatment has a high-level ionic strength caused by many variables.

Many modern ideas about complexation description in concentrated solutions are based on Pitzer representations [30,31]. According to [32], the value of the mean activity coefficient in the aqueous salt solution decreases by 10–15% and then increases. The work [33] confirmed the tendency for the activity coefficient to change with increasing salt concentration. Mathematical models of Pitzer and Bromley were used in the calculations. The activity coefficients of REE presented in the literature can hardly be called reliable since they were estimated in a specific system and conditions. There is no existing data on the activity of complex REE ions, in particular, carbonate and phosphate.

Technologically important thermodynamic and kinetic parameters of dissolution and precipitation are practically absent in real systems. The research [34] carried out the process of dissolving rare earth metal carbonates in ammonium carbonate in the presence of ammonium carbonate solution. The authors provided the resulting data and confirmed that the method has a sufficiently high recovery rate of REE and it is easy to scale. However, the study does not indicate what happens to the concentration of REE in the presence of an alkaline ammonia solution. In the study [35], the process of dissolving lanthanide carbonate was considered as neodymium carbonate in a sodium carbonate solution. A model of metastable dissolution of the solid phase was proposed, in which each layer splits and enters into a complexation reaction, then enters the solution. At the time of the study, there was no data on the equilibrium composition of the aqueous phase as well as the activity coefficients of all possible REE complexes in the carbonate–alkaline system. In this regard, it was decided to build a mathematical model using the found equilibrium constants.

Nevertheless, there is still no solution on how to explain the imperfection of the solution at high ionic strength. There is no information in the literature on a detailed description of the solubility of REE carbonates in alkaline carbonate media.

Therefore, the purpose of this study is to obtain and analyze isotherms of the solubility of REE in carbonate solution to compile a predictive mathematical model describing the behavior of lanthanides in the system.

## 2. Materials and Methods

Dry powders of REE carbonates have been synthesized. For this, chemically pure nitrates of rare earth metals were dissolved in distilled water. Further, the addition of the carbonate solution produced precipitates. The precipitates were dried and crushed. Using the methods of X-ray powder diffractometry and X-ray fluorescence analysis, the composition and structure of the obtained powders were confirmed. The particle size of powders in the range of 40–80 microns was determined. A solution of chemically pure potassium carbonate was poured into the batch reactor (HEL), and sediments of rare-earth metals were added. The following parameters were monitored during dissolution: mixing rate; temperature; and pH of the solution. The conditions for the experimental studies are shown in Table 1.

The authors [36] point out the applicability of complexonometric analysis to determine the concentration of REE. The property of rare earth metals to form strong complexes with arsenazo (III) was used. Stability constants of carbonate complexes from [20] are used to describe equilibria in aqueous solutions, Table 2. The constant $\beta_{1}^{0}$ applies to the LnL complex, and $\beta_{2}^{0}$ to the $LnL_{2}$ complex, where L is a ligand.

## 3. Results

As a result of the experiment, solubility isotherms were obtained. Figure 1 shows the dependence of the REE concentration on the carbonate ion in the solution.

With an increase in the concentration of carbonate ion, an increase in the concentration of lanthanide in the aqueous phase is observed. This can be explained by an increase in the effect of lanthanide complexation. However, when a certain concentration of carbonate ion is reached, the concentration of lanthanide ceases to grow. In the study [37], when describing the kinetics of the dissolution process, it was already indicated that when the temperature increases, the solubility isotherm curves have a sharper slope, and a horizontal plateau is reached at a lower carbonate ion concentration. This indicates an acceleration of the process and indicates an endothermic reaction. 

## 4. Discussion

The bicarbonate REE complex is the most stable [21,22,28]; thus, the general reaction of dissolving the carbonate form of lanthanide to obtain carbonate complexes of the composition is as follows:(7)$${Ln}_{2}{\left(CO_{3}\right)}_{3}+\left(n-1\right)M_{2}CO_{3}=2M_{\left(2n-3\right)}\left[{Ln\left(CO_{3}\right)}_{n}\right],$$
where M—$Na,\;K,\;NH_{4}$.

The reaction (7) is influenced by the carbonate ion concentration, the pH of the medium, the nature of the REE, and the temperature. In this system, along with heterogeneous equilibrium, complexation processes coexist, which also affects the dissolution process of the sparingly soluble lanthanide compound. The high ionic strength of the solution should also be taken into account, and, as a result, low coefficients of ion activity.

The equilibrium constant for reaction (7), according to the law of mass action, will be as follows:(8)$$K=\frac{\alpha^{2}\left({Ln(CO_{3})}_{2}^{-}\right)}{\alpha \left(CO_{3}^{2-}\right)}=\frac{{\left[{Ln(CO_{3})}_{2}^{-}\right]}^{2}}{\left[CO_{3}^{2-}\right]}\times \frac{\gamma^{2}\left({Ln(CO_{3})}_{2}^{-}\right)}{\gamma \left(CO_{3}^{2-}\right)}=Q\times \Pi_{\gamma }$$
where Q—concentration constant of the dissolution process; α—activity; γ—activity coefficient, and $\Pi_{\gamma }$—parameter that takes into account the coefficients of ion activity.

To find the equilibrium constant, the dependence was interpolated by zero ionic strength Q = f(F), where F is the function of ionic strength by Debye–Hückel theory:(9)$$F=\frac{\sqrt{I}}{1+\sqrt{I}}$$
where I—ionic strength.

The parameter $\Pi_{\gamma }$ at zero ionic strength is equal to one. The ionic strength of the solution is calculated using the following formula:(10)$$I=0.5\times \sum C_{i}\times z_{i}^{2}$$
where $C_{i}$—ion concentration, mol/kg, and $z_{i}$ is the charge of the ion. The ionic strength in the carbonate–alkaline solution was calculated using the following formula:(11)$$I=0.5\times \left(C_{M_{2}CO_{3}}\times 1\times 2^{2}+C_{M_{2}CO_{3}}\times 2\times 1^{2}\right)=3C_{M_{2}CO_{3}}$$
where $C_{M_{2}CO_{3}}$—concentration of potassium carbonate solution, mol/kg.

Using experimental data, equilibrium constants were obtained for four REE in carbonate–alkaline medium, as shown in Table 3.

Comparing the equilibrium constants, it can be seen that with the high ionic strength of the carbonate solution, the solubility of the solid phase increases significantly due to complexation.

In order to describe the activity coefficients, several approaches were considered by Davies, Bromley, and [38,39,40,41] to the assessment of activity coefficients. The authors [39,40] state that the third Debye–Hückel approximation is sufficient to calculate the activity coefficient at high ionic strength. The activity coefficients calculated in this way are identical to the data in [28,41]. The carbonate ion activity factor at ionic strength up to 2.5 M can be determined according to [40] with sufficient accuracy by the following formula:(12)$$log\gamma_{CO_{3}^{2-}}=-\frac{Az^{2}\sqrt{I}}{1+Ba\sqrt{I}}$$
where z is the charge of the ion, A, B and a are constants in the Debye–Hückel equation; A = 0.504; B = 3.237 × 10^−9^; a = 4.5 × 10^8^.

The dependence of the carbonate complex REE on the ionic strength is a more complex function; the calculation method is reflected in the work [39]. The activity coefficient of the bicarbonate REE complex in the solution can be calculated using the simplified Formula (13) and the expanded Formula (14):(13)$$log\gamma_{{Ln(CO_{3})}_{2}^{-}}=\frac{\alpha_{1}\sqrt{I}}{1+\sqrt{I}}$$
(14)$$log\gamma_{{Ln(CO_{3})}_{2}^{-}}=\frac{\alpha_{0}\sqrt{I}}{1+\sqrt{I}}-\alpha_{3}\times I-\alpha_{4}\times I^{3/2}$$

A semi-empirical model based on the Debye–Hückel equation was used to calculate the activity coefficients. The coefficients α_0_–α_4_ are different for different REE and are selected in a mathematical model until the greatest reliability is achieved. The formula for calculating the equilibrium concentration of the REE complex in the aqueous phase is obtained from the law of mass action (8):(15)$$\left[{Ln(CO_{3})}_{2}^{-}\right]=\frac{-2K+\sqrt{4K^{2}+4\Pi_{\gamma }KC_{M_{2}CO_{3}}}}{2\Pi_{\gamma }}$$
where K is equilibrium constant for reaction (7), and $C_{M_{2}CO_{3}}$ is the concentration of metal carbonate solution.

The mathematical model best describes the activity coefficient using the calculated Formula (14). Therefore, this model was selected. The built solubility isotherms based on the calculated and experimental data are shown in Figure 2.

The data obtained from the calculation are consistent with the experiment. Increasing carbonate solution concentration naturally increases the solubility of REE carbonates. In order to check the applicability of the mathematical model, the water–salt composition was calculated. Using the REE concentration at equilibrium in the aqueous phase, the composition of REE complexes can be calculated. Equilibrations in the solution of complex compounds can be described based on mole fractions of individual complexes that have a relationship with the free ligand content in the solution. The total concentration of the complexing ion is the following expression:(16)$$C_{Me}=\left[Me\right]+\left[MeL\right]+\left[{MeL}_{2}\right]+\ldots +\left[{MeL}_{n}\right]$$
where $\left[Me\right]$—concentration of the noncomplexed ions; $\left[{MeL}_{n}\right]$—concentration of complexes.

Using the stability constants of the complexes in the formula, rewrite the expression (16) as follows:(17)$$C_{Me}=\left[Me\right]+\beta_{1}\left[Me\right]\left[L\right]+\beta_{2}\left[Me\right]{\left[L\right]}^{2}+\ldots +\beta_{n}\left[Me\right]{\left[L\right]}^{n}=\left[Me\right]\left(1+\sum_{i}^{n}{\beta_{i}\left[L\right]}^{i}\right),$$
where β—concentration constants of stability of complexes, and $\left[L\right]$—ligand concentration.

Equilibrium concentrations of complexes can be calculated as follows:(18)$$\left[MeL_{n}\right]=C_{Me}\alpha_{MeL_{n}}$$
where $\alpha_{MeL_{n}}$—mole fraction of the complex.

Molar fractions of REE complexes that can be formed in a carbonate–alkaline solution are calculated using the following formula:(19)$$\alpha_{MeL_{n}}=\frac{\left[MeL_{n}\right]}{C_{Me}}=\frac{{\beta_{n}\left[L\right]}^{n}}{1+\sum_{i}^{n}{\beta_{i}\left[L\right]}^{i}}=\alpha_{Me}{\beta_{n}\left[L\right]}^{n}$$
where $\alpha_{Me}$—mole fraction of the noncomplexed ion.

In the system under consideration, the role of metal is played by the trivalent lanthanide ion ${Ln}^{3+}$. Taking into account the reactions (1)–(5) taking place in the solution and using Formula (17), we obtain the following formula for calculating the total concentration of complexing lanthanide:(20)$$\begin{array}{c} C_{Me}=\left[{Ln}^{3+}\right]+\beta_{{LnCO}_{3}^{+}}\left[{Ln}^{3+}\right]\left[CO_{3}^{2-}\right]+\beta_{{Ln(CO_{3})}_{2}^{-}}\left[{Ln}^{3+}\right]{\left[CO_{3}^{2-}\right]}^{2}+\\ \beta_{{Ln(HCO_{3})}^{2+}}\left[{Ln}^{3+}\right]\left[HCO_{3}^{-}\right]+\beta_{LnOH^{2+}}\left[{Ln}^{3+}\right]\left[OH^{-}\right]+\beta_{Ln(O{H)}_{2}^{+}}\left[{Ln}^{3+}\right]{\left[OH^{-}\right]}^{2} \end{array}$$
where $\beta_{{LnCO}_{3}^{+}}$—stability constant of monocarbonate complex; $\beta_{{Ln(CO_{3})}_{2}^{-}}$—bicarbonate complex stability constant; $\beta_{{Ln(HCO_{3})}^{2+}}$—stability constant of lanthanide hydrocarbonate; $\beta_{LnOH^{2+}}$—stability constant of the first stage hydroxo-complex, $\beta_{Ln(O{H)}_{2}^{+}}$—stability constant of the second stage hydroxo-complex.

The ionic strength value of 0.68 mol/kg was taken for calculation since the concentration constants of stability of lanthanide complexes are presented in the works [30,36]. The calculated equilibrium composition of the forms of lanthanide in the aqueous phase at an ionic strength of 0.68 mol/kg is shown in Table 4.

Since the bicarbonate REE complex is the most stable in the carbonate–alkaline environment, almost the entire volume of lanthanide will be associated with this configuration, which is confirmed by calculations. The adequacy of the mathematical model is confirmed by an accurate interpretation of experimental data with an error of less than 5%. The obtained constants and coefficients are shown in Table 5.

The data obtained were used to plot the dependence of the activity coefficients on the ionic strength of the solution (Figure 3).

The concentrations of REE complexes in the aqueous phase calculated using a mathematical model are shown in Figure 4. The figure also shows the dependence of the sum of the concentration of ${LnCO}_{3}^{+}$ and ${Ln(HCO_{3})}^{2+}$ on the ionic strength.

Comparing the equilibrium constants obtained in the experiment with the thermodynamic data [20], significant differences can be noted. The equilibrium constant of reaction (6) is the smallest, while reaction (7) has the highest equilibrium constant. This is due to the increase in the solubility of the solid phase of lanthanide carbonate during complex formation. The bicarbonate complex $\left[{Ln(CO_{3})}_{2}^{-}\right]$ plays the most active part in the complex formation reaction. This is consistent with the highest stability constant of the complex (Table 2). At low ionic strengths with the bicarbonate complex, there are also complexes ${LnCO}_{3}^{+}$ and ${Ln(HCO_{3})}^{2+}$ (Figure 4), the concentration of which rapidly decreases with increasing ionic strength of the carbonate solution. The calculated activity coefficients (Figure 3) present proper results in the calculation of thermodynamic equilibrium. However, this calculation method can only be applied to the carbonate–alkaline solution.

## 5. Conclusions

The carbonate–alkali system presented in this work is complex and multicomponent due to the formation of carbonate complexes. When REE carbonates are dissolved at a high ionic strength achieved by potassium carbonate, the formation of water-soluble lanthanide complexes is observed. After a system analysis, it was determined that the formation of a bicarbonate complex of REE is thermodynamically more beneficial than other complexes.

Solubility isotherms of sparingly soluble carbonates were constructed, and the following equilibrium constants logK at zero ionic strength were calculated: Nd −11.3; Sm −8.6; Gd −8.0; and Ho −7.3. The experimentally received equilibrium constants were used to build a predictive mathematical model. During the analysis of isotherms, the coefficients of the expanded Debye–Hückel equation were calculated. The activity coefficients of the bicarbonate complexes of neodymium, samarium, gadolinium, and holmium were estimated. The difference in the obtained data is due to the difference in REE. The increase in the value of the equilibrium constant in Nd-Sm-Gd-Ho can be explained by two trends. There is a decrease in the radius of the trivalent ion of rare earth metals, as well as a transition from a group of light to heavy REE. Based on the available concentration constants of stability of the complexes, the water–salt composition was considered. The adequacy and applicability of the predictive mathematical model can be confirmed by its high accuracy in comparing solubility isotherms with experimental data.

## Figures and Tables

**Figure 1 materials-16-03140-f001:**
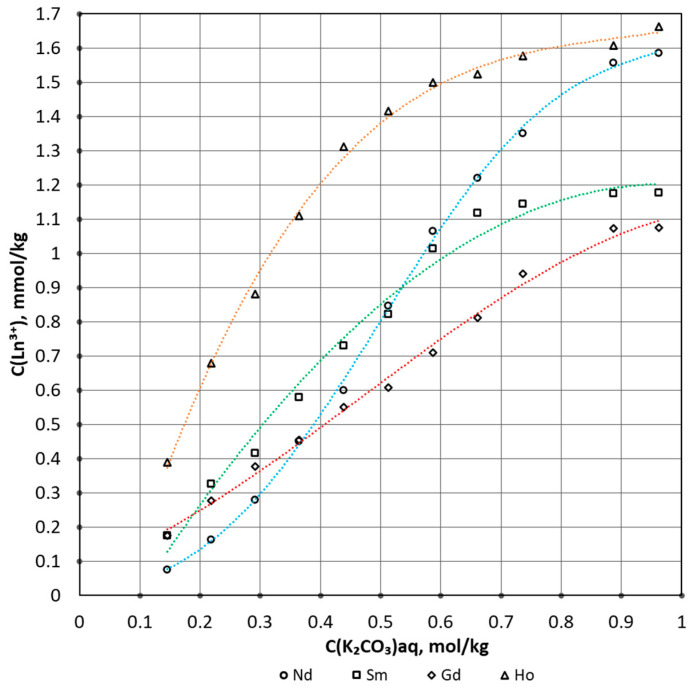
Solubility isotherms: blue line for neodymium, green line for samarium, red line for gadolinium, orange line for holmium.

**Figure 2 materials-16-03140-f002:**
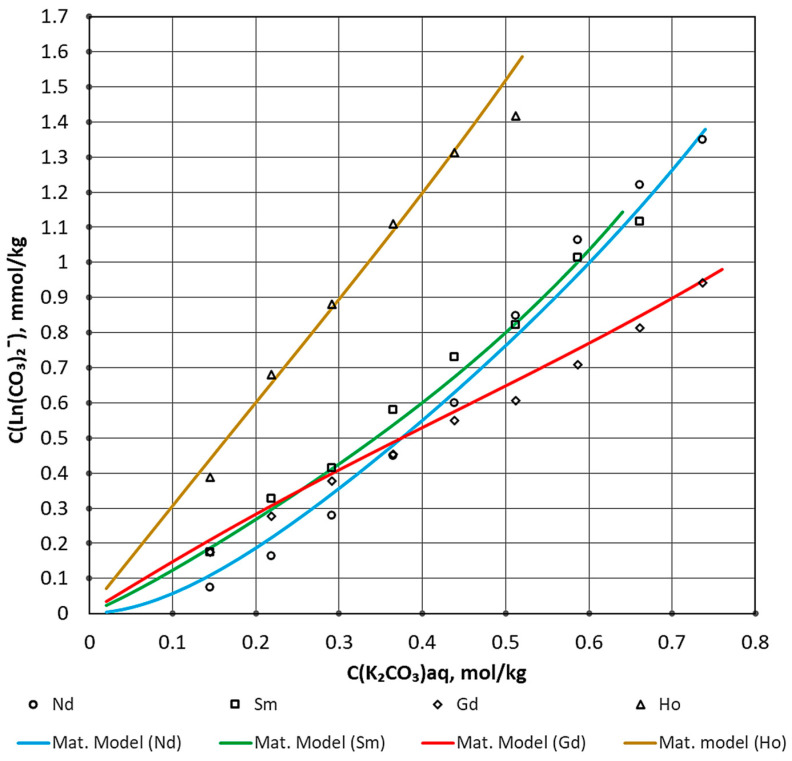
Isotherm of solubility for REE in carbonate–alkaline medium.

**Figure 3 materials-16-03140-f003:**
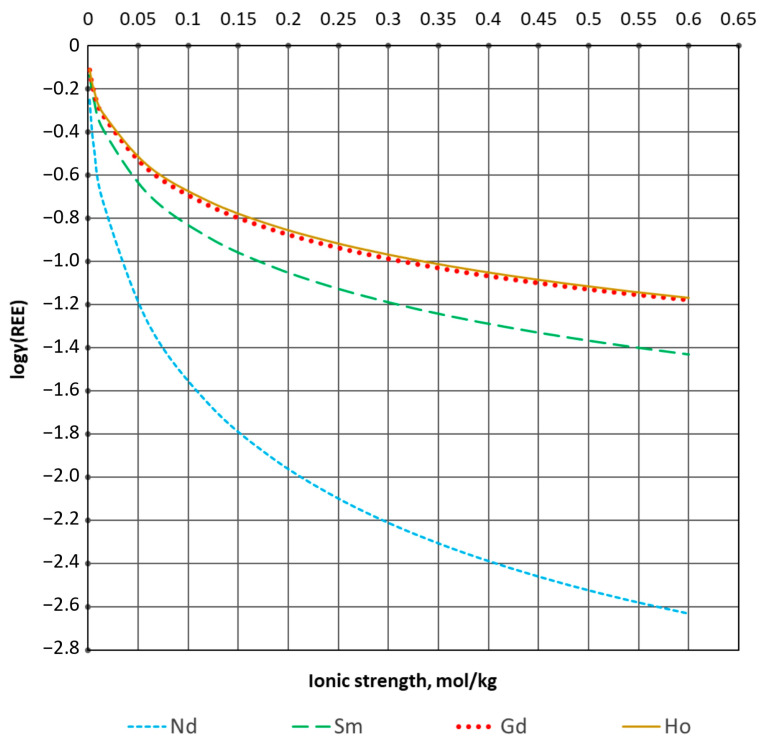
REE activity coefficients in carbonate–alkaline solution.

**Figure 4 materials-16-03140-f004:**
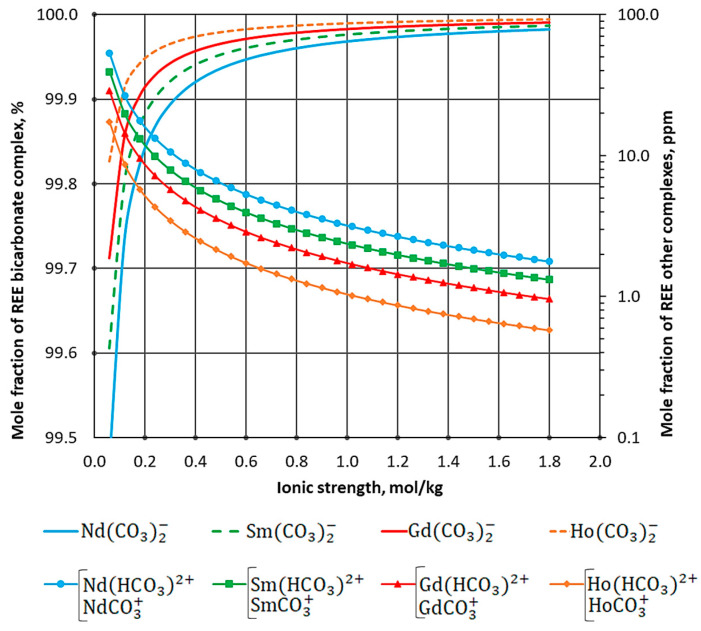
Dependence of concentration of REE complexes on ionic strength.

**Table 1 materials-16-03140-t001:** Parameters of sediment carbonation of REE.

Process Parameter	Value
Concentration ${CO}_{3}^{2-}$ in solution, mol/kg	0.145–1.116
Ionic strength I, mol/kg	0.436–3.347
Agitation intensity, rpm	1000
Temperature, K	298
Mixing time, min	50
Equilibrium period, h	24
pH	8.0–12.5
Ratio l:s, mL/g	2100

**Table 2 materials-16-03140-t002:** Stability constants.

Cation	Ligand	$\beta_{1}^{0}$	$\beta_{2}^{0}$
Nd^3+^	$CO_{3}^{2-}$	7.6	12.6
Nd^3+^	$HCO_{3}^{-}$	2.12	-
Nd^3+^	$OH^{-}$	5.9	11.1
Sm^3+^	$CO_{3}^{2-}$	7.8	12.8
Sm^3+^	$HCO_{3}^{-}$	2.1	-
Sm^3+^	$OH^{-}$	6.1	11.5
Gd^3+^	$CO_{3}^{2-}$	7.8	13.1
Gd^3+^	$HCO_{3}^{-}$	2.1	-
Gd^3+^	$OH^{-}$	6.0	11.8
Ho^3+^	$CO_{3}^{2-}$	8.0	13.3
Ho^3+^	$HCO_{3}^{-}$	2.17	-
Ho^3+^	$OH^{-}$	6.1	11.9

**Table 3 materials-16-03140-t003:** Equilibrium constants in carbonate–alkaline medium.

Cation	logK for Reaction (6)	logK_exp_	I, mol/kg
Nd^3+^	−34.6	−11.3	0.4–2.7
Sm^3+^	−34.5	−8.6	0.4–2.2
Gd^3+^	−34.7	−8.0	0.4–2.7
Ho^3+^	−33.8	−7.3	0.4–2.0

**Table 4 materials-16-03140-t004:** Part of bicarbonate REE complex.

Element	Mole Fraction χ			
	${Ln}^{3+}$, ppm	${LnCO}_{3}^{+}$, %	${Ln(CO_{3})}_{2}^{-}$, %	${Ln(HCO_{3})}^{2+}$, ppm
Nd^3+^	0.005	0.047	99.953	0.019
Sm^3+^	0.003	0.035	99.965	0.007
Gd^3+^	0.001	0.025	99.975	0.003
Ho^3+^	0.001	0.015	99.985	0.001

**Table 5 materials-16-03140-t005:** Modeled data.

Element	logK_exp_	α_0_	α_1_	α_3_	α_4_
Nd	−11.3	−6.7	−5.6	−0.74	0.28
Sm	−8.6	−3.6	−3.3	−0.40	0.22
Gd	−8.0	−3.0	−2.6	−0.33	0.14
Ho	−7.3	−2.9	−2.6	−0.27	0.14

## Data Availability

Not applicable.

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
