# Peer review of "Complex Formation of Rare-Earth Elements in Carbonate–Alkaline Media"

_materials, 2023, doi:10.3390/ma16083140_

Round 1
Reviewer 1 Report
General observations/suggestions-
· The continuity in the manuscript especially in section 2 (i.e., Materials and Methods) and section 3 (i.e. results) should be improved.
· While citing references in manuscript, provide more explanation on methodology adopted by the authors (for reference no.5,6,7,8,9)
· The English writing can be improved in section 1 (i.e. Introduction)
Specific queries-
Line 16-18 |
The statement repeated twice. Kindly correct it. |
Line 69-72 |
Refrain the statement with more explanation on Solubility of Yttrium and Lanthanide compounds |
Line 80-83 |
Kindly elaborate the statement and simplify it. |
Line 96-98 |
Check if any grammatical error and explain the same. |
Line 109-113 |
Simplify/refrain the statement |
Equation 8 |
Is “α” stands for mole fraction or activity coefficients? Mention it |
Equation 9 |
Does “I” stands for ionic strength or anything else, kindly elaborate on it. |
Line 174-177 |
Kindly check what “Q” stands for? Is it concentration constant or zero ionic strength, kindly check it |
Line 191-192 |
Elaborate on “carbonate ion activity factor at ionic strength up to 2,5M”. |
Equation 11 |
What do z,A,B,a stand for in equation 11? |
Equation 14 |
Explain Equation 14 and what do K and C stand for? |
Author Response
Thank you for the detailed and thorough analysis of the manuscript. I would like to answer your questions in the attached file.

Reviewer 2 Report
The manuscript describes solubility products for carbonate salts of Nd, Sm, Gd, and Ho. The solubilty products were acculately evaluated with stability constants of various species and activity coefficients. However, following issues are included in the manuscript. Those should be modified before the publication.
1. Introduction
The section contains a lot of information. However, the too much information make difficult to understand significant contents such as issues, and motivation. The author should make a point and decrease complicated information.
2. Technologically important thermodynamic and kinetic parameters
What are the technologically important thermodynamic and kinetic parameters? Technologically important points should be explained.
3. Digits in the data of equilibrium constants and activity coefficients.
The values show up to three digits after decimal points. Can the solublity experiments give such accurate values? The authors described the the expermental data contain the error of less than 5%. This error should be included the values.
4. Disscussion of equilibrium contants and activity coefficients.
The authors need to compare the values of equilibrium contants or solubility products for carbonate salts of the rare earth elements with previously reported data for those salts and discuss differences of parameters, included reactions, activity coefficients, and so on. The discussion make us be understood importance and merits of the data.
5. Check of manuscript
Several mistakes were found. The authors need to overall check the manuscript.
L. 69; solubility insoluble
L. 80; the most signicant are
L. 166; Me in the eq. (7)
Author Response

(The authors gave the same response as above.)

Reviewer 3 Report
Complex Formation of Rare-Earth Elements in Carbonate-Alkaline Media is very interesting paper. Some improvement is required.
Line 13: equilibrium of carbonate-alkali systems of rare earth metals. What is reason for this choice? Why not sulfate-acidic or chloride acidic?
Line 115, 116: The authors provided resulted data and confirm that the method has a sufficiently high recovery rate of REE. What is high recovery rate of REE (95%?)
Line 256, 257:After a system analysis, it was determined that the formation of a bicarbonate complex of REE is thermodynamically more beneficial than other complexes. Which type of complexes?
Author Response
Thank you for the detailed and thorough analysis of the article. I will try to answer your questions in the attached file.

Reviewer 4 Report
Materials 2313460
In this paper, the authors show the study of the formation of rare-earth elements in carbonate-alkaline media. Rare earth metals are critical component for many industries. The extraction of these materials presents many problems. The authors present a study on the formation and equilibrium of carbonate alkali systems and evaluate the values of equilibrium constants. A mathematical model was developed, which allows to calculate the water salt composition.
The introduction is good and the references appropriate. Is necessary that the authors compare the results and values obtained for different metals.
Comment
1) Define Ln and REE. Which is the difference?
2) Revise the presentation of the Figure 1
3) Revise the formula (10)
4) Comment with major detail the Figure 2
5) Complete the conclusions
Author Response

(The authors gave the same response as above.)

Round 2
Reviewer 2 Report
The authors basically did not modify based on my suggestions. I recommend reject of this paper if the authors do not make the corrections.
1. The section of introduction is too long. This paper is not reveiew article. Introduction should be brief.
3. The obtained thermodynamic parameters has 5% errors. There are scientifically no meanings to show up to three digits after decimal points.
4. The authors need to compare the values of equilibrium contants or solubility products for carbonate salts of the rare earth elements with previously reported data for those salts and discuss differences of parameters, included reactions, activity coefficients, and so on. The discussion make us be understood importance and merits of the data. Present manuscript has no discussion, and therfore should change article types to notes.
Author Response
Thank you for the detailed and thorough analysis of the manuscript. I would like to answer your questions. All the changes are in the text of the manuscript.
